# Discriminating miRNA Profiles between Endometrioid Well- and Poorly-Differentiated Tumours and Endometrioid and Serous Subtypes of Endometrial Cancers

**DOI:** 10.3390/ijms21176071

**Published:** 2020-08-23

**Authors:** Lenka Kalinkova, Karol Kajo, Miloslav Karhanek, Lenka Wachsmannova, Peter Suran, Iveta Zmetakova, Ivana Fridrichova

**Affiliations:** 1Department of Genetics, Cancer Research Institute, Biomedical Research Center of Slovak Academy of Sciences, 845 05 Bratislava, Slovakia; Lenka.Kalinkova@savba.sk (L.K.); Lenka.Wachsmannova@savba.sk (L.W.); Iveta.Zmetakova@savba.sk (I.Z.); 2Department of Pathology, St. Elisabeth Cancer Institute, 812 50 Bratislava, Slovakia; karol.kajo@ousa.sk (K.K.); peter.suran@ousa.sk (P.S.); 3Laboratory of Bioinformatics, Biomedical Research Center of Slovak Academy of Sciences, 845 05 Bratislava, Slovakia; Miloslav.Karhanek@savba.sk

**Keywords:** miRNA expression, endometrioid endometrial carcinoma, serous endometrial carcinoma, molecular classification

## Abstract

The discrimination of different subtypes of endometrial carcinoma (EC) is frequently problematic when using the current histomorphological classification; therefore, new markers for this differentiation are needed. Here, we examined differences in miRNA expression between well- and poorly-differentiated (grades 1 and 3) endometrioid endometrial carcinoma (EEC) and between EEC and serous endometrial carcinoma (SEC). The expression of 84 tumour-suppressor miRNAs was analysed by real-time polymerase chain reactions in 62 EC and 20 non-neoplastic endometrial specimens. The potential functions of the differentially expressed miRNAs were determined by bioinformatics analyses. The expression of let-7c-5p, miR-125b-5p, miR-23b-3p, and miR-99a-5p in grade 3 EEC was decreased compared to grade 1 EEC. To discriminate between EEC and SEC, let-7g-5p, miR-195-5p, miR-34a-5p, and miR-497-5p expression was significantly downregulated in SEC. In bioinformatic analyses, miRNAs that could discriminate grade 1 from grade 3 mainly targeted genes involved in PI3K-AKT signaling, whereas miRNAs that could discriminate EEC from SEC targeted genes involved in several signaling pathways, but mainly MAPK signaling. Taken collectively, our results indicate that the activation of certain signaling pathways can be useful in the molecular characterization of EEC and SEC.

## 1. Introduction

Endometrial carcinoma (EC) is one of the most common gynaecological malignancies and it associates with increasing mortality. According to the Global Cancer Statistics Study from 2018, uterine cancer represents 37% of all reproductive malignancies and accounts for 21.2% of all cancer deaths [1]. Approximately 75% of individuals are diagnosed with EC at early stages (FIGO (International Federation of Gynecology and Obstetrics) I and II) and the 5-year overall survival rate of these individuals ranges from 74% to 91% compared to 66% and 26% for those diagnosed at late stages (FIGO III and IV) [2,3].

Bokhman’s classification is based on the clinical, metabolic, and endocrinological characteristics of EC [4]. Type I (70–80% of all ECs) represents endometrioid endometrial carcinoma (EEC) with good or moderate differentiation, good prognosis, and hormone receptor overexpression. This type of EC develops from atypical hyperplasia, and the risk factors are endogenous oestrogens, nulliparity, and obesity. Type II (10–20% of all ECs) represents serous endometrial carcinoma (SEC) with poor differentiation, as well as other types, namely clear cell EC, high-grade EEC, and other undifferentiated/de-differentiated carcinomas with mesenchymal phenotypes. This type of EC is common in older women without hormonal or metabolic changes, and it develops from an atrophic endometrium and premalignant lesions. The prognosis of type II EC is poor [5]. On the other hand, the current World Health Organization classification is based on the cytomorphological characteristics of EC [6], although these characteristics do not provide sufficient information on tumour behaviour. Moreover, the classification of ECs with ambiguous morphologies is problematic.

For these reasons, it is important to identify other biological characteristics to predict tumour behaviour and prognosis. This is also important for treatment because patients are diagnosed according to type, grade, and stage of the disease. Generally, low-grade EEC is diagnosed at early stages and treated with radiotherapy, whereas high-grade EEC, SEC, clear cell EC, and carcinosarcomas are often diagnosed at late stages and treated additionally with adjuvant chemotherapy [7]. Although the phenotypic features of tumours reflect the molecular changes occurring during disease progression, the most promising approach for cancer management is the identification of biomarkers that could further sub-classify the tumours.

In 2012, the Cancer Genome Atlas reported frequent mutations in *PTEN*, *CTNNB1*, *PIK3CA*, *ARID1A*, *KRAS*, and *ARID5B*, as well as rare *TP53* mutations and copy number changes in most types of EECs. A subgroup of EECs manifested hot-spot *POLE* mutations and microsatellite instability (MSI), whereas, 25% of high-grade EECs and SECs showed frequent mutations in *TP53*, extensive changes in copy number, DNA methylation, and low hormone receptor expression. These findings resulted in the establishment of a new EC classification comprised of four categories, namely *POLE* ultra-mutated, MSI hyper-mutated, low copy-number, and high copy-number [6,7,8,9]. Although mutations have been identified in different genes, these mutated genes are not restricted to a particular subtype [3,8,10]. Therefore, the criteria for the precise discrimination of different EC subtypes have not been fulfilled.

Aberrant DNA hypermethylation is an important epigenetic event in the development and progression of EC. Hypermethylation of *MLH1*, *RASSF1A/2A*, *PTEN*, and *APC* has been reported in EEC and less frequently in SEC [11,12,13,14], which may be a consequence of the increased expression of DNA methyltransferases DNMT1 and DNMT3B in EEC [15]. Furthermore, hypermethylation of *HOXA10* in EC and *ADCYAP1* and *HAND2* in benign lesions may increase the risk of EC [16,17]. In comparative whole-genome analyses of the DNA methylome, 27,009 and 15,676 differentially methylated regions were identified in EEC and SEC, respectively, indicating the complex and distinct regulation of promoters and enhancers in these two subtypes of EC [18]. 

Two decades ago, a new mechanism of epigenetic regulation involving protein non-coding RNA molecules, microRNAs (miRNAs), was described. Aberrant expression of miRNAs can contribute to the development and progression of many types of cancer. Furthermore, miRNAs can function as oncogenes, tumour suppressors, modulators of metastatic dissemination, and regulators of cancer stem cells. Therefore, they may represent a new class of biomarkers in the diagnosis and treatment of EC [19]. In a database that summarized miRNA expression profiles in 184 different types of cancer (http://mircancer.ecu.edu, updated on 31 October 2019), only 71 miRNAs associated with EC compared with lung, colorectal, and breast cancers with 815, 636, and 572 miRNAs, respectively [20]. A recent meta-analysis of 74 original articles and eight literature reviews identified 261 miRNAs involved in EC, including 133 oncogenic miRNAs, 110 miRNAs with tumour-suppressor activities, and 18 miRNAs with ambiguous functions [21].

Several studies have reported correlations between abnormal miRNA expression and pathological characteristics. In a study of 141 EC patients, significant differences in miRNA expression were observed between EC and the normal endometrium, as well as in early EC (FIGO stage 1A and grade 1). In advanced EC, miR-199c overexpression was a predictor of better prognosis [22]. In another study, decreased expression of miR-101, miR-10b*, miR-139-5p, miR-152, miR-29b, and miR-455-5p associated with a worse prognosis of SEC [23].

To identify new markers in order to improve the early detection of EC, the roles of 37 miRNAs in EC were evaluated by bioinformatics analyses. Using in silico analyses, 12 miRNAs were identified, which targeted genes belonging to three groups. Group 1 genes regulated cell survival by activating *EGFR*, *PTEN*, *PIK3CA*, *ILK*, *AKT3*, and *FOXO3*. Group 2 genes regulated cell proliferation and growth by activating *SOS1*, *KRAS*, *RAF1*, *MAP2K3*, *GSK3b*, *AXIN2*, *LEF1,* and *CCND1*. Finally, group 3 genes regulated the cell cycle, DNA repair, and apoptosis by activating *TP53* [24].

In this retrospective study, we identified four miRNAs that were downregulated in poorly differentiated (grade 3) EEC. We also identified another four miRNAs for discriminating EEC from SEC, which were significantly downregulated in SEC. Using bioinformatic analyses, we identified the potential functions of these miRNAs. We found that miRNAs discriminating grade 1 EEC from grade 3 EEC targeted several genes in the PI3K-AKT pathway, whereas miRNAs discriminating EEC from SEC targeted genes involved in several signaling pathways, but mainly the MAPK pathway.

## 2. Results

A set of 84 miRNAs showing tumour suppressor functions in different types of cancer was used for miRNA profiling in well- and poorly-differentiated EEC and SEC. The analyses were performed in 62 EC patients and 20 control individuals and there was no statistically significant difference in age between populations (*t*-test, *p* = 0.91).

### 2.1. Differences in miRNA Expression between Well- and Poorly-Differentiated EEC

There were significant differences in 36 miRNAs between grade 1 EEC and grade 3 EEC and controls by miRNA profiling (Table 1), in which five genes were upregulated and 27 genes were downregulated in grade 1 and six genes were upregulated and 30 genes were downregulated in grade 3 compared to controls (Figure 1A). Of these, let-7c-5p (fold change 0.4866, *p* = 0.002877), miR-125b-5p (fold change 0.4694, *p* = 0.012155), miR-23b-3p (fold change 0.4037, *p* = 0.002478), and miR-99a-5p (fold change 0.4804, *p* = 0.011175) were significantly downregulated in grade 3 EEC compared to grade 1 EEC. The normalized miRNA expression levels represented by 2^−Δ*C*t^ for let-7c-5p, miR-125b-5p, miR-23b-3p, and miR-99a-5p were 0.514 and 0.250, 4.428 and 2.078, 1.863 and 0.752, and 0.827 and 0.397 for grades 1 and 3, respectively, compared to 1.455, 16.846, 5.279, and 3.301 for controls, respectively. Differentially expressed miRNAs in grades 1 and 3 and controls are summarized in Figure 2A.

### 2.2. Differences in miRNA Expression between Endometrioid and Serous EC

There were differences in 35 miRNAs between EEC and SEC by miRNA profiling (Table 1), in which six genes were upregulated and 26 genes were downregulated in EEC and six genes were upregulated and 29 genes were downregulated in SEC compared to controls (Figure 1B). Of these, miRNAs let-7g-5p (fold change 0.4133, *p* = 0.004899), miR-195-5p (fold change 0.4829, *p* = 0.022248), miR-34a-5p (fold change 0.367, *p* = 0.000961), and miR-497-5p (fold change 0.2508, *p* = 0.00005) were significantly downregulated in SEC compared to EEC. The normalized miRNA expression levels were 0.643 and 0.266 for let-7g-5p, 1.350 and 0.652 for miR-195-5p, 0.428 and 0.157 for miR-34a-5p, and 0.187 and 0.047 for miR-497-5p in EEC and SEC, respectively, compared to 1.260, 3.421, 0.593, and 0.552 for controls (Figure 2B).

Furthermore, different EC-specific miRNA expression profiles were evidenced for miR-145-5b in the comparison of the two EEC grades, for miR-143-5p and miR-424-5p in the comparison of EEC and SEC, and for miR-1-3p in both comparisons. Compared to controls, the levels of all miRNAs were downregulated, although the differences were not significant. Furthermore, miR-137 was downregulated in grade 1 but upregulated in grade 3, whereas miR-1-3p was downregulated in EC regardless of the de-differentiation level and histological subtype.

### 2.3. Aberrant miRNA Expression in EC Tumourigenesis

To investigate the roles of the differentially expressed miRNAs in endometrial tumourigenesis, we identified the target genes of all significantly downregulated miRNAs discriminating grade 1 from grade 3 EEC and EEC from SEC using the reverse search module in the miRPath software. Target genes, as well as their *p*-values, are summarized in Table 2. For several miRNA–target interactions, the strong evidence was experimentally confirmed by reporter assays, Western blotting, or qPCR (Table 2, Appendix A).

Furthermore, we analyzed 44 target genes with respect to the eight discriminating miRNAs. We found that these genes could be divided into three distinct groups according to the criteria for discriminating between grades 1 and 3 or EEC and SEC or both (Table 3). For the first group, miR-125b-5p and miR-23b-3p potentially regulated seven genes. For the second group, different miRNAs identified in both comparisons potentially regulated 19 target genes. For the third group, four miRNAs potentially regulated 18 genes (Table 3). miRNA–target gene interactions are graphically depicted in Figure 3.

To confirm the involvement of the 44 target genes in EC, we used another bioinformatics tool, David 6.7. In this analysis, the 44 target genes were found to be involved in EC, but also in prostate, colorectal, non-small lung, or acute myeloid cancers (Appendix A). The Pathview software was used to identify the genes involved in different signaling pathways (Figure 4). 

Five out of the top seven genes in the first group, which were regulated by miRNAs discriminating grade 1 EEC from grade 3 EEC (Table 3), were involved in PI3K-AKT signaling. However, miRNAs discriminating EEC from SEC were involved in several signaling pathways, but mainly in MAPK signaling (Figure 4).

## 3. Discussion

Accurate therapeutic management of patients with EC depends on comprehensive clinical and pathological evaluations. Histomorphological examinations are critical in the diagnosis of EC, but the classification of ECs by histomorphological criteria has limited reproducibility [25]. More precise tools for distinguishing of these tumours are needed, which should facilitate subtype-specific research and treatment of EC [26].

Cell differentiation is the meaningful process resulting in well-defined tissues and organs, which is in cancer cells usually disrupted. The pathological grading of malignant tumours represents the extent of reversion of the differentiated phenotype; therefore, the score inversely correlates with the extent of cell differentiation. The multifunctional tumour suppressor p53 has roles in the cell cycle, apoptosis, senescence, DNA repair, and differentiation [27,28], although its role in de-differentiation is unknown. In EC, *TP53* is frequently mutated; however, the high p53 immunohistochemical staining simultaneously with *TP53* point mutations were found [29]. Regarding the molecular variability, it is difficult to interpret the p53 immunohistochemical results [30]. A previous study reported that miR-125b-5p targets *TP53* [31]. In this study, a downregulation in the miR-125b-5p level in grade 3 EEC suggested that *TP53* expression may increase, thereby affecting the process of cell differentiation. A downregulation in the miR-34a-5p level in SEC affected *TP53* expression and could induce frequent de-differentiation of serous cancer cells [32].

Members of let-7 family are aberrantly expressed in many cancers and a correlation between weak let-7 expression and poor differentiation of aggressive tumours was observed [33]. Consistent with our results, a previous study has reported decreased let-7c-5p and miR-99a-3p expression in early stage EEC [34], as well as decreased let-7c-5p expression in grade 3 [35]. The PI3K/AKT/mTOR signaling pathway regulates cancer cell proliferation, migration, survival, and angiogenesis, and *AKT1* and *mTOR* genes have been identified as targets of miR-99a-3p in other types of cancers [36,37]. Compared to controls, miR-99a-3p expression was significantly decreased and negatively correlated with EEC tumour grade [38,39]. In another study, decreased miR-125b expression increased the expression of *ERBB2*, its target gene [40]. ERBB2/HER2 overexpression was identified in very aggressive primary tumours and metastatic lesions of EC, mostly in the serous subtype, suggesting that these changes induce AKT phosphorylation [40,41]. Histologically, among the poorly differentiated ECs, SEC predominantly shows nuclear atypia and papillary growth compared to EEC. Grade 3 EEC is characterized by solid glandular growth and less frequent papillary formation with transition from one to another component. These features allow clinicians to discriminate between endometrioid and serous EC, but in many patients, reproducible classification has not been possible [42].

Of the four discriminating miRNAs between EEC and SEC identified in this study, only miR-34a-5p has been studied extensively. In non-selective EC, a decreased miR-34a-5p level was reported to associate with higher stage and metastasis [43,44], whereas another study revealed that miR-34a-5p targets *SNAIL* and other genes such as those involved in epithelial-mesenchymal transition (EMT) [45]. In another study, decreased miR-34a-5p expression upregulated the *L1CAM* level in EC cell lines and tumours [46]. In stage I EC patients, *L1CAM* expression was found to be a strong independent predictor of disease recurrence and poor prognosis [47]. On the other hand, increased miR-34a-5p expression was found to decrease the expression of its target genes *NOTCH1* and *DLL1* in EC. NOTCH1 and DLL1 are regulators of cell proliferation, differentiation, and apoptosis; therefore, these expression changes contribute to EC [48]. For other miRNAs, such as miR-195 and miR-497, changes in expression have been reported in SEC [23], but further studies are needed.

There is only one published comparative study of miRNA expression profiles in 14 and nine samples of EEC and SEC, respectively. Of 667 miRNAs, only 17 were significantly discriminating between these two EC subtypes. Of these, let-7c, miR-9, miR-146a, miR-450a, miR-504, and miR-542-3p showed at least a 2-fold difference in expression between EEC and SEC [49]. In this study, we identified another four miRNAs downregulated in EEC, whose levels decreased even further in SEC. The higher expression of their target genes in a more aggressive subtype of SEC indicate the important roles of these miRNAs in EC progression.

In endometrial cancer cells, molecular changes in PI3K/AKT/mTOR, MAPK/ERK, WNT/β-catenin, VEGF/VEGFR, ERBB, P53/P21, and P16INK4a/pRB signaling pathways are responsible for evading apoptosis, increasing cell proliferation, inhibiting differentiation, and stimulating angiogenesis [50]. For instance, PI3K/AKT signaling is critical for EMT and the stemness phenotype in endometrioid and non-endometrioid EC, and these changes have been reported to be mediated by several miRNAs that target *PTEN*, *TWIST1*, *ZEB1*, and *BMI1* [51]. 

There is little knowledge on the roles of specific miRNAs in EC regarding their aberrant miRNA expression, target genes, and functions in EC tumourigenesis; therefore, to understand the clinical significance of our results, we performed bioinformatic analyses. To differentiate between grades 1 and 3, we identified let-7c-5p, miR-125b-5p, miR-23b-3p, and miR-99a-5p, as well as seven in silico generated target genes involved in EC progression. These target genes were regulated by miR-125b-5p or miR-23b-3p, and five are members of the PI3K/AKT signaling pathway, including phosphatidylinositol 3-kinases (*PIK3CD*, *PIK3R3*), serine/threonine kinases (*PDPK1*, *AKT1*), and the transcription factor *FOXO3*. Similarly, let-7g-5p, miR-195-5p, miR-34a-5p, and miR-497-5p, which differentiated EEC from SEC, regulated target genes participating in PI3K/AKT signaling. These target genes are also members of the WNT signaling pathway. They can also contribute to microsatellite instability by targeting *MLH1* and exclusively influence MAPK signaling by regulating cell proliferation and growth [50]. Furthermore, overexpression of growth factors in cancers can constitutively and autonomously activate MAPK signaling [52]. It is possible that these miRNAs can affect the expression of *GRB2*, *SOS2*, *KRAS*, *ARAF*, *MAPK2K1*, *MAPK1*, and *ELK1*. For instance, KRAS is a small GTPase that transduces signals between cell surface receptors and the nucleus. Mutations in *KRAS* were identified in 20–30% of EEC cases compared to 3% of SECs [9], suggesting that the regulation of *KRAS* by let-7g-5p, miR-195-5p, or miR-497-5p could be another mechanism contributing to SEC.

Cancer development and progression involves the deregulation of essential cellular processes. miRNAs regulate the expression of multiple genes; however, it is challenging to characterize all the consequences of these changes. Here, we identified two sets of downregulated miRNAs and two key signaling pathways that could be used to differentiate between grade 1 and grade 3 EEC and between EEC and SEC. The associations between activated proteins of these pathways and EC cell phenotypes or behaviours need to be widely investigated; however, our findings may be helpful in the differential diagnosis, especially in the classification of morphologically mixed or problematic EC. 

## 4. Materials and Methods

### 4.1. Patients

In this retrospective study, we analyzed endometrial tumour tissues isolated from FFPE specimens from histomorphologically-characterized EC. In brief, 20 and 21 specimens were from patients with grade 1 EEC and grade 3 EEC, respectively; and 21 specimens were from patients with SEC. The age of all 62 EC patients ranged from 45 to 86 years, with a mean 65.18 ± 8.85 years (63.75, 63.76, and 67.95 years for EEC grade 1, EEC grade 3, and SEC groups, respectively). We also included 20 non-neoplastic endometrial specimens from women in matched age from 53 to 79 years (mean, 65.4 ± 6.75 years). All specimens were obtained from the tissue archives of the St. Elizabeth Cancer Institute, Bratislava, Slovakia, and prepared before 2018. The St. Elizabeth Cancer Institute Review Board approved this retrospective study and waived the consent of patients and control persons. No patient underwent preoperative radiotherapy or chemotherapy before specimen collection. Controls had no signs or symptoms of cancer or other serious diseases.

### 4.2. miRNA Extraction and Real-Time PCR

miRNAs from FFPE endometrial tumour tissues were isolated using the miRNeasy FFPE Kit (Qiagen, Hilden, Germany) according to the manufacturer’s instructions. miRNA samples of suitable purity were reversely transcribed into cDNA using the miScript II RT Kit (Qiagen) according to the manufacturer’s instructions. For real-time polymerase chain reactions, we used the miScript miRNA PCR Array Human Tumor Suppressor miRNAs Kit containing 84 known tumour suppressor miRNAs in different cancer types (MIHS-119Z, Qiagen) using the miScript SYBR Green PCR Kit (Qiagen). RT-PCR reactions were carried out in an AriaMx Real-Time PCR System (Agilent, Santa Clara, CA, USA) using conditions as follows: Predenaturation at 95 °C for 15 min, followed by 40 cycles at 94 °C for 15 s, 55 °C at 30 s, and 70 °C for 30 s.

### 4.3. Statistical Analysis

Real-time PCR data was statistically analyzed using the miScript miRNA PCR Array Data Analysis Software (Qiagen). Global normalization was performed using six endogenous small RNAs, namely SNORD61, SNORD68, SNORD72, SNORD95, SNORS96A, and RNU6-6P. For the quantification of relative gene expression, the ΔΔCT method was used [53]. Fold-change 2^−ΔΔ*C*t^ was the normalized gene expression 2^−Δ*C*t^ in the test sample divided by the normalized gene expression 2^−Δ*C*t^ in the control sample. The upregulation of miRNA expression was indicated by fold-change values greater than 2, and the fold-regulation was equal to the fold-change. By contrast, the downregulation of miRNA expression was indicated by fold-change values less than 0.5, and the fold-regulation was the negative inverse of the fold-change (–1/x). *p*-Values were calculated based on the Student’s *t*-test of replicate 2^−Δ*C*t^ values for each gene in both groups, and statistical significance was considered at *p* < 0.05. In bioinformatic analyses, the reverse-search module of the DIANA-miRPath v3.0 online software suite was used to identify the most relevant target genes involved in EC for eight discriminating miRNAs [54]. A reverse-search was performed separately against every available database included in miRPath such as Tarbase v.7.0, TargetScan v.6, and microT-CDS v.5.0. For the independent confirmation of the identified target genes with respect to pathways involved in EC and other cancer types, the online software tool David 6.7 was used [55]. For network visualization of miRNA–gene interactions, Cytoscape v.3.8.0 was used [56]. Experimentally validated miRNA–target interactions were adopted from miRTarBase [57]. For pathway analysis, the KEGG Database and Pathview Tool were used [58]. Heat maps were generated using the OriginPro 2015 software (OriginLab Corp., Northampton, MA, USA).

## Figures and Tables

**Figure 1 ijms-21-06071-f001:**
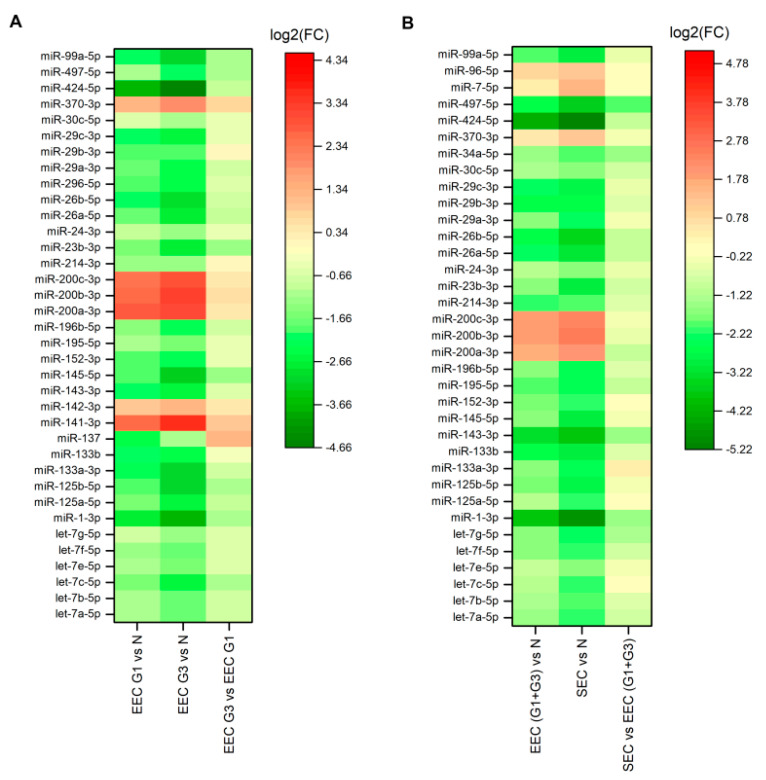
Heat maps of miRNA expression in endometrial cancer. Differences in miRNA expression between grade 1 (EEC G1) and grade 3 (EEC G3) endometrioid endometrial carcinoma compared with non-neoplastic endometrium (N), (**A**) and between endometrioid endometrial carcinoma (EEC (G1 + G3)) and serous endometrial carcinoma (SEC) compared with the non-neoplastic endometrium (N) (**B**). On the scale of fold regulation (log2(FC)), red corresponds to upregulated miRNAs and green corresponds to downregulated miRNAs.

**Figure 2 ijms-21-06071-f002:**
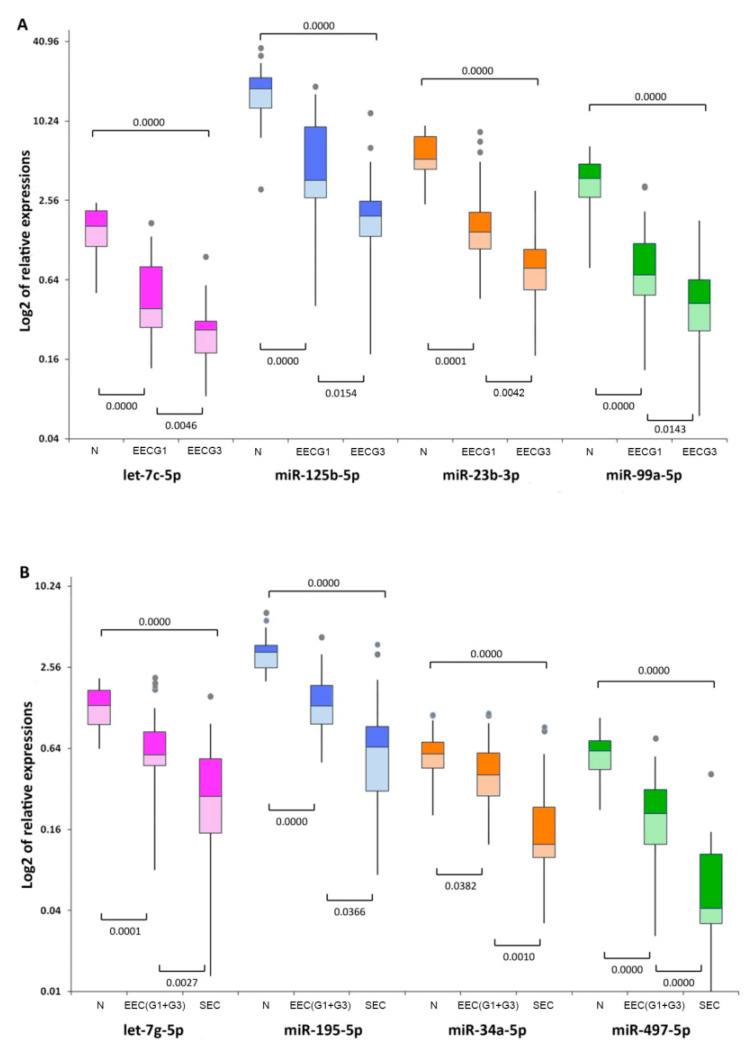
Relative expression of differentially expressed miRNAs. Differential expression of let-7c-5p, miR-125b-5p, miR-23b-3p, and miR-99a-5p between grade 1 (EEC G1) and grade 3 (EEC G3) endometrioid endometrial carcinoma compared with the non-neoplastic endometrium (N) (**A**), and differential expression of let-7g-5p, miR-195-5p, miR-34a-5p, and miR-497-5p between endometrioid endometrial carcinoma (EEC(G1 + G3)) and serous endometrial carcinoma (SEC) compared with the non-neoplastic endometrium (N) (**B**). Box lengths correspond to the interquartile range (IQR) and represent values between 75th and 25th percentiles. Values greater than 1.5 IQRs from the end of the box are labeled as outliers (grey points). Horizontal lines depict the medians. Statistical significance was considered at *p* < 0.05.

**Figure 3 ijms-21-06071-f003:**
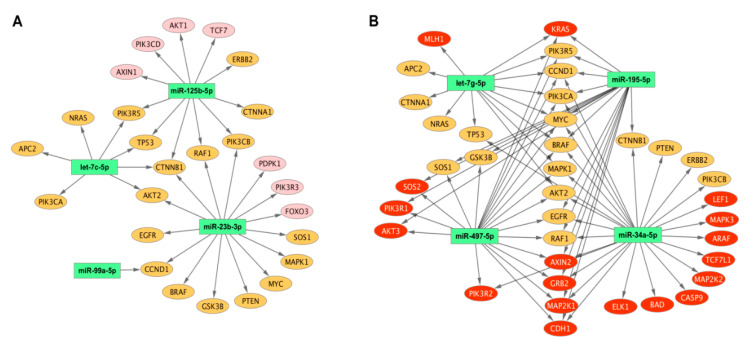
MicroRNA–target gene network for discriminating miRNAs. MicroRNA–target gene network for miRNAs discriminating grade 1 from grade 3 (**A**) and endometrioid from serous endometrial carcinoma (**B**). Target genes regulated by miRNAs discriminating between different grades in endometrioid and endometrioid and serous carcinoma are depicted in pink and red, respectively. Target genes regulated by miRNAs from both sets are depicted in orange. The associations between microRNAs and target genes are represented by black arrows. The scheme was generated by CytoScape 3.8.0.

**Figure 4 ijms-21-06071-f004:**
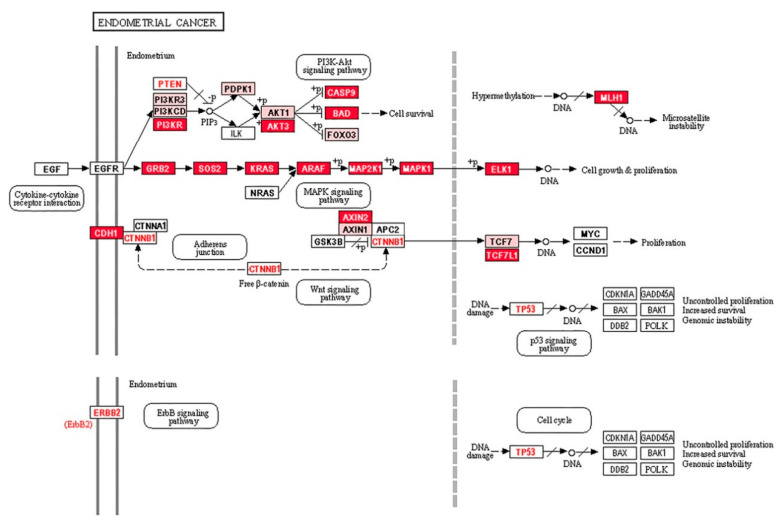
Signaling pathways involved in EC. Pink boxes represent the first group of target genes, which are targeted by two miRNAs, miR-125b-5p and miR-23b-3p that can discriminate grade 1 from grade 3. Red boxes represent the third group of target genes for miRNAs discriminating serous from endometrioid endometrial samples. *KRAS*, *CTNNB1*, *ERBB2*, *PTEN*, and *TP53* (letters in red, except *KRAS* for visibility) are frequently mutated, whereas *CTNNB1*, *ERBB2*, *PTEN*, and *TP53* are regulated by both sets of discriminating miRNAs (second group of target genes).

**Table 1 ijms-21-06071-t001:** Differentially expressed miRNAs in EC subtypes.

EC Subtypes	Differently Expressed miRNAs in Each of Two Evaluated Groups Compared to Controls
Endometrioid grade 1 vs. grade 3	let-7a-5p, let-7b-5p, let-7c-5p, let-7e-5p, let-7f-5p, let-7g-5p, miR-1-3p, miR-125a-5p, miR-125b-5p, miR-133a-3p, miR-133b, miR-137, miR-141-3p, miR-142-3p, miR-143-3p, miR-145-5p, miR-152-3p, miR-195-5p, miR-196b-5p, miR-200a-3p, miR-200b-3p, miR-200c-3p, miR-214-3p, miR-23b-3p, miR-24-3p, miR-26a-5p, miR-26b-5p, miR-296-5p, miR-29a-3p, miR-29b-3p, miR-29c-3p, miR-30c-5p, miR-370-3p, miR-424-5p, miR-497-5p, miR-99a-5p
Endometrioid vs. Serous	let-7a-5p, let-7b-5p, let-7c-5p, let-7e-5p, let-7f-5p, let-7g-5p, miR-1-3p, miR-125a-5p, miR-125b-5p, miR-133a-3p, miR-133b, miR-143-3p, miR-145-5p, miR-152-3p, miR-195-5p, miR-196b-5p, miR-200a-3p, miR-200b-3p, miR-200c-3p, miR-214-3p, miR-23b-3p, miR-24-3p, miR-26a-5p, miR-26b-5p, miR-29a-3p, miR-29b-3p, miR-29c-3p, miR-30c-5p, miR-34a-5p, miR-370-3p, miR-424-5p, miR-497-5p, miR-7-5p, miR-96-5p, miR-99a-5p

Underlined miRNAs were significantly differentially expressed between grade 1 and 3 endometrioid endometrial carcinoma (EC) and between endometrioid and serous EC.

**Table 2 ijms-21-06071-t002:** Target genes of significantly downregulated miRNAs in more advanced subtypes of endometrial carcinoma identified by bioinformatic analysis.

EC Subtypes	Downregulated miRNAs *	Database	Enrichment *p*-value	Target Genes
Endometrioid grade 1 vs. grade 3	let-7c-5p	TargetScan	1.80 × 10^20^	*NRAS*, *PIK3R5*, *TP53*, *AKT2*, *CCND1*, *APC2*, *PIK3CA*
		microT-CDS	5.74 × 10^3^	*NRAS*
	miR-125b-5p	Tarbase	1.80 × 10^20^	*ERBB2*,*RAF1*, *AXIN1*,*TP53*, *CTNNB1*, *CTNNA1*,*AKT1*
		TargetScan	3.60 × 10^11^	*PIK3CB*, *PIK3R5*,*PIK3CD*,*TCF7*
		microT-CDS	5.74 × 10^03^	*PIK3CB*
	miR-23b-3p	Tarbase	6.13 × 10^34^	*RAF1*, *EGFR*, *AKT2*, *CCND1*, *CTNNB1*, *MYC*, *SOS1*, *FOXO3*, *PDPK1*,*PTEN*, *MAPK1*
		TargetScan	2.79 × 10^17^	*GSK3B*, *PIK3R3*, *CCND1*, *SOS1*, *PDPK1*, *PTEN*
		microT-CDS	3.54 × 10^14^	*BRAF*, *PIK3CB*, *PIK3R3*, *PDPK1*, *PTEN*
	miR-99a-5p	Tarbase	5.74 × 10^3^	*CCND1*
Endometrioid vs. Serous	let-7g-5p	Tarbase	4.49 × 10^27^	*BRAF*, *NRAS*,*KRAS*, *MLH1*, *TP53*, *CCND1*, *CTNNA1*,*MYC*, *MAPK1*
		TargetScan	1.80 × 10^20^	*NRAS*, *PIK3R5*, *TP53*,*AKT2*, *CCND1*, *APC2*, *PIK3CA*
		microT-CDS	5.74 × 10^3^	*NRAS*
	miR-195-5p	Tarbase	1.80 × 10^20^	*CDH1*,*CCND1*, *CTNNB1*, *MYC*, *AKT3*, *PIK3CA*, *GRB2*
		TargetScan	4.89 × 10^41^	*GSK3B*, *SOS2*, *PIK3R5*,*RAF1*, *EGFR*, *KRAS*, *CDH1*,*CCND1*, *AXIN2*, *PIK3R1*, *SOS1*, *AKT3*, *MAP2K1*
		microT-CDS	9.75 × 10^24^	*BRAF*, *SOS2*,*RAF1*,*CCND1*, *AXIN2*, *PIK3R1*, *AKT3*, *MAP2K1*
	miR-34a-5p	Tarbase	1.11 × 10^83^	*BRAF*, *MAP2K2*, *PIK3R2*, *TCF7L1*, *RAF1*, *EGFR*, *ARAF*,*TP53*, *AKT2*, *CDH1*,*CCND1*, *CTNNB1*,*AXIN2*,*MYC*, *MAPK3*, *CASP9*, *PIK3CA*,*LEF1*,*MAP2K1*, *PTEN*, *MAPK1*, *GRB2*, *BAD*, *ELK1*
		TargetScan	2.82 × 10^8^	*CCND1*,*LEF1*,*MAP2K1*
		microT-CDS	3.54 × 10^14^	*ERBB2*, *PIK3CB*,*AXIN2*,*LEF1*,*MAP2K1*
	miR-497-5p	Tarbase	1.78 × 10^30^	*SOS2*, *PIK3R2*, *AKT2*, *CCND1*, *MYC*, *AKT3*, *PIK3CA*,*MAP2K1*, *MAPK1*, *GRB2*
		TargetScan	4.89 × 10^41^	*GSK3B*, *SOS2*, *PIK3R5*,*RAF1*, *EGFR*, *KRAS*, *CDH1*, *CCND1*, *AXIN2*, *PIK3R1*, *SOS1*, *AKT3*, *MAP2K1*
		microT-CDS	9.75 × 10^24^	*BRAF*, *GSK3B*, *SOS2*,*RAF1*, *AXIN2*, *PIK3R1*, *AKT3*,*MAP2K1*

* miRNAs were significantly downregulated in both subtypes compared to controls. For underlined target genes, the strong evidence for miRNA targeting was experimentally confirmed by reporter assays, Western blotting, or qPCR (data from miRTarBase).

**Table 3 ijms-21-06071-t003:** List of 44 target genes identified by the miRPath software and divided into three distinct groups with respect to discriminating miRNAs for both comparisons of endometrial carcinomas.

	Endometrioid Grade 1 vs. Grade 3				Endometrioid vs. Serous				
Target Genes	let-7c-5p	miR-125b-5p	miR-23b-3p	miR-99a-5p	let-7g-5p	miR-195-5p	miR-34a-5p	miR-497-5p	Number of Targeting miRNAs
*FOXO3*			Yes						1
*PIK3R3*			Yes						1
*AXIN1*		Yes							1
*PIK3CD*		Yes							1
*TCF7*		Yes							1
*AKT1*		Yes							1
*PDPK1*			Yes						1
*APC2*	Yes				Yes				2
*NRAS*	Yes				Yes				2
*CTNNA1*		Yes			Yes				2
*ERBB2*		Yes					Yes		2
*PTEN*			Yes				Yes		2
*GSK3B*			Yes			Yes		Yes	3
*PIK3CB*		Yes	Yes				Yes		3
*SOS1*			Yes			Yes		Yes	3
*EGFR*			Yes			Yes	Yes	Yes	4
*MAPK1*			Yes		Yes		Yes	Yes	4
*TP53*	Yes	Yes			Yes		Yes		4
*AKT2*	Yes		Yes		Yes		Yes	Yes	5
*CTNNB1*	Yes	Yes	Yes			Yes	Yes		5
*MYC*			Yes		Yes	Yes	Yes	Yes	5
*RAF1*		Yes	Yes			Yes	Yes	Yes	5
*BRAF*			Yes		Yes	Yes	Yes	Yes	5
*PIK3R5*	Yes	Yes			Yes	Yes		Yes	5
*PIK3CA*	Yes				Yes	Yes	Yes	Yes	5
*CCND1*			Yes	Yes	Yes	Yes	Yes	Yes	6
*CDH1*						Yes	Yes	Yes	3
*MAP2K1*						Yes	Yes	Yes	3
*GRB2*						Yes	Yes	Yes	3
*KRAS*					Yes	Yes		Yes	3
*AXIN2*						Yes	Yes	Yes	3
*SOS2*						Yes		Yes	2
*PIK3R2*							Yes	Yes	2
*AKT3*						Yes		Yes	2
*PIK3R1*						Yes		Yes	2
*MAP2K2*							Yes		1
*TCF7L1*							Yes		1
*ARAF*							Yes		1
*MAPK3*							Yes		1
*CASP9*							Yes		1
*LEF1*							Yes		1
*BAD*							Yes		1
*ELK1*							Yes		1
*MLH1*					Yes				1

Genes targeted by the eight discriminating miRNAs that associate with EC are included (DIANA-miRPath v3.0 reverse-search module).

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
