# Peer review of "Discriminating miRNA Profiles between Endometrioid Well- and Poorly-Differentiated Tumours and Endometrioid and Serous Subtypes of Endometrial Cancers"

_ijms, 2020, doi:10.3390/ijms21176071_

Round 1
Reviewer 1 Report
Dear Authors, I read with great interest the paper, although I have some concerns.
First, you have to check the English and rephrase several sentences. For instance the first line of the paragraph is not in english, I can understand the meaning but it must be rephrase.
Second, please answer to the following corrections
- Page 2 line 33 -34, add study on HOXA hypermethylation, ref https://doi.org/10.3109/01443615.2013.776027
- Page 9 line 7-10. Split the sentence in two: it's too long.
- Page 9 line 12: please add reference for the first sentence of the discussion. Also the sentence is too strong, you can not say that the management of endometrial cancer is always unsuccessful. I suggest to report data on recurrences in apparently low risk patients.
- Page 10 line 15, add the following reference on PI3K https://doi.org/10.3892/or.2018.6939
- Page 10 line 46, please clarify "strictly histomorphologically-characterized EC"
Author Response
Response to Reviewer 1 Comments
Thank you reviewer for recommendations, which help to improve our manuscript.
The changes are highlighted by turquoise color.
Point 1: Page 2 line 33 -34, add study on HOXA hypermethylation, ref https://doi.org/10.3109/01443615.2013.776027.
Response 1. New reference [16] of paper published by Fambrini et al. from University of Florence was included to the manuscript. The references were re-numbered.
Point 2: Page 9 line 7-10. Split the sentence in two: it's too long.
Response 2: The long sentence was changed as:
“Five out of the top seven genes in the first group, which were regulated by miRNAs discriminating grade 1 EEC from grade 3 EEC (Table 3), were involved in PI3K-AKT signalling. However, miRNAs discriminating EEC from SEC were involved in several signalling pathways, but mainly in MAPK signalling (Figure 4).”
Point 3: Page 9 line 12: please add reference for the first sentence of the discussion. Also the sentence is too strong, you can not say that the management of endometrial cancer is always unsuccessful. I suggest to report data on recurrences in apparently low risk patients.
Response 3: We agree with the reviewer's opinion that the sentence originally presented “The improper characterization of endometrial tumours classified solely according to their histomorphological characteristics results in unsuccessful therapeutic management.” does not correctly capture the principle, and we therefore propose a change to:
“Accurate therapeutic management of patients with EC depends on comprehensive clinical and pathological evaluations. Histomorphological examinations are critical in the diagnosis of EC, but the classification of ECs by histomorphological criteria has limited reproducibility [25]. More precise tools for distinguishing of these tumours are needed, which should facilitate subtype-specific research and treatment of EC [26].”
Two new references [25 and 26] were involved and the list of references was re-numbered.
Comment to Point 3:
It´s true, that low-risk EC has a minimal risk for pelvic lymph node metastasis (≤ 5%), vaginal recurrence (1-3%), and pulmonary metastases (< 1%) (Güngördük et al, 2018; Chi et al, 2008), but some patients (2-15%) with well-differentiated EEC in the early stages of the disease and without other risk signs are (in the new genomic / molecular classification) included in the p53 molecular group within low-grade EEC and their inclusion in this molecular category is associated with unfavorable course of the disease (Stelloo et al, 2016; Talhouk et al, 2015; Talhouk et al, 2017; Kommoss et al, 2018; Britton et al, 2019; Yano et al, 2019; Vermij et al, 2020).
References:
• Güngördük K., Firat Cüylan Z, Kahramanoglu I., et al. Risk Factors for Recurrence in Low-Risk Endometrial Cancer: A Case-Control Study. Oncol Res Treat 2018;41:466-470.
• Chi DS, Barakat RR, Palayekar MJ, et al. The incidence of pelvic lymph node metastasis by FIGO staging for patients with adequately surgically staged endometrial adenocarcinoma of endometrioid histology. Int J Gynecol Cancer 2008;18:269-273.
• Stelloo E, Nout RA, Osse EM et al. Improved risk assessment by integrating molecular and clinicopathological factors in early stage endometrial cancer-combined analysis of the portec cohorts. Clin. Cancer Res. 2016; 22; 4215–4224.
• Talhouk A, McConechy MK, Leung S et al. A clinically applicable molecular-based classification for endometrial cancers. Br. J. Cancer 2015; 113; 299–310.
• Talhouk A, McConechy MK, Leung S et al. Confirmation of promise: a simple, genomics-based clinical classifier for endometrial cancer. Cancer 2017; 123; 802–813.
• Kommoss S, McConechy MK, Kommoss F et al. Final validation of the promise molecular classifier for endometrial carcinoma in a large population-based case series. Ann. Oncol. 2018; 29; 1180–1188.
• Britton H, Huang L, Lum A et al. Molecular classification defines outcomes and opportunities in young women with endometrial carcinoma. Gynecol. Oncol. 2019; 153; 487– 495.
• Yano M, Ito K, Yabuno A et al. Impact of tp53 immunohistochemistry on the histological grading system for endometrial endometrioid carcinoma. Mod. Pathol. 2019; 32; 1023–1031.
• Vermij L, Smit V, Nout R, Bosse T. Incorporation of molecular characteristics into endometrial cancer management. Histopathology. 2020;76:52-63.
Point 4: Page 10 line 15, add the following reference on PI3K https://doi.org/10.3892/or.2018.6939
Response 4: Recommended paper published by Malentacchi et al. from University of Florence was not included; therefore, it is out of concept of this manuscript. In recommended paper, the changes in PI3K pathway in grade 3 EC were presented resulting from PTEN and PIK3CA mutations, not through miRNA regulation of these genes that is the main topic of our manuscript.
Point 5: Page 10 line 46, please clarify "strictly histomorphologically-characterized EC"
Response 5: In presented study, selection of EC patients was based only on histological typing and grading that means histomorphological features, without the evaluation of staging parameters and without the use of IHC markers, e.g. PTEN, p53, ER, PR, etc., so we used the term "strictly", but we acknowledge your comment and omit the term "strictly".
Note to English changes:
Firstly submitted version of our manuscript was professionally edited by International Science editing (www.internationalscienceediting.com) as was mentioned in Cover letter to editor. The same company secondly edited this revised version.
Reviewer 2 Report
Lenka Kalinkova and co-authors identified miRNAs that differ among various grades of endometrial cancer. Results are very clearly presented and convincing. Methodology was chosen correctly, bioinformatic analysis was also conducted accurately. The entire approach is modern and can be adapted to other tumor types in serach for possible markers or targets for pharmacological interventions.
The only one concern I have is the lack of experimental validation of miRNA targets. Page 6 line 16 - "We analysed 44 target genes with respect to the eight discriminating miRNAs." - I tried to figure out if data in Table 3 show experimental results or again bioinformatic analysis. It is not clear for reader how this part of the entire story was approached. Some miRNA targets, ideally regulated by the single miRNA as indicated in Table 3, should be quantified and their relative expression between Endometrioid grade 1 vs grade 3 and Endometrioid vs Serous should be included in the table.
Author Response
Thank you reviewer for comment, which helps to improve our manuscript.
The changes are highlighted by yellow color.
Point 1: The only one concern I have is the lack of experimental validation of miRNA targets. Page 6 line 16 - "We analysed 44 target genes with respect to the eight discriminating miRNAs." - I tried to figure out if data in Table 3 show experimental results or again bioinformatic analysis. It is not clear for reader how this part of the entire story was approached. Some miRNA targets, ideally regulated by the single miRNA as indicated in Table 3, should be quantified and their relative expression between Endometrioid grade 1 vs grade 3 and Endometrioid vs Serous should be included in the table.
Response 1. For our study, only limited quantity of FFPE samples from selected EC subtypes were available, which were used for miRNA expression analyses. For evaluation of potential functions of changed miRNA expressions and clinical importance of our findings, we performed bioinformatic analyses using several softwares and databases. Regarding this, we modified sentence (previously page 6 line 16) and legend of Table 3. as:
“Furthermore, we analysed 44 target genes with respect to the eight discriminating miRNAs. We found that these genes could be divided into three distinct groups according to the criteria for discriminating between grades 1 and 3 or EEC and SEC or both (Table 3).”
“Table 3. List of 44 target genes identified by miRPath Software and divided into three distinct groups with respect to discriminating miRNAs for both comparisons of endometrial carcinomas.”
Round 2
Reviewer 2 Report
I do not have any further comments.